# The Baltic Earth Assessment Reports

H. E. Markus Meier[1], Marcus Reckermann[2], Joakim Langner[3], Ben Smith[4,5] and Ira Didenkulova[6]

[1]Department of Physical Oceanography and Instrumentation, Leibniz Institute for Baltic Sea Research Warnemünde, Rostock, Germany
[2]International Baltic Earth Secretariat, Helmholtz-Zentrum Hereon, Geesthacht, Germany
[3]Swedish Meteorological and Hydrological Institute, Norrköping, Sweden
[4]Hawkesbury Institute for the Environment, Western Sydney University, Australia
[5]Department of Physical Geography and Ecosystem Science, Lund University, Sweden
[6]Department of Mathematics, University of Oslo, Norway

*Correspondence to*: H. E. Markus Meier (markus.meier@io-warnemuende.de)

**Abstract.** Baltic Earth is an independent research network of scientists from all Baltic Sea countries that promotes regional Earth system research. Within the framework of this network, the Baltic Earth Assessment Reports (BEARs) were produced in the period 2019-2022. These are a collection of 10 review articles summarising current knowledge on the environmental and climatic state of the Earth system in the Baltic Sea region and its changes in the past (palaeoclimate), present (historical period with instrumental observations) and prospective future (until 2100) caused by natural variability, climate change and other human activities. The division of topics among articles follows the grand challenges and selected themes of the Baltic Earth Science Plan, such as the regional water, biogeochemical and carbon cycles, extremes and natural hazards, sea level dynamics and coastal erosion, marine ecosystems, coupled Earth system models, scenario simulations for the regional atmosphere and the Baltic Sea, and climate change and impacts of human use. Each review article contains an introduction, the current state of knowledge, knowledge gaps, conclusions and key messages, based on which recommendations for future research are made. Based on the BEARs, Baltic Earth has published an information leaflet on climate change in the Baltic Sea as part of its outreach work, which has been published in two languages so far, and organised conferences and workshops for stakeholders, in collaboration with the Baltic Marine Environment Protection Commission (HELCOM).

## 1 Introduction

### 1.1 BALTEX/Baltic Earth history

Baltic Earth[1] is an international research network dealing with Earth system sciences in the Baltic Sea region (Fig. 1). It is politically independent and focuses on research on the water and energy cycles, climate variability and climate change, water management and extreme events, and associated impacts on marine and terrestrial biogeochemical cycles. Research on human impact on the Earth system in more general terms, i.e. the anthroposphere, defined as the part of the environment created or modified by humans for use by human activities, was also included in the Baltic Earth Science Plan (2017)[2].

Baltic Earth is the successor of the Baltic Sea Experiment (BALTEX) programme, which was founded in 1993 as a GEWEX continental-scale experiment (Global Energy and Water Exchanges, a core project of the World Climate Research Programme) (Reckermann et al., 2011). In the first phase (1993–2002), BALTEX was primarily devoted to hydrological, meteorological and oceanographic processes in the Baltic Sea drainage basin and thus focused on physical aspects of the Earth system. In the second phase (2003–2012), the programme was expanded to include regional climate research, biogeochemical cycles including carbon, engagement with stakeholders and decision-makers via assessment reports, as well as communication and education, i.e. the organisation of summer and winter schools and international master courses.

---

[1] https://baltic.earth, last access: 4 February 2023

[2] https://baltic.earth/grandchallenges, last access: 4 February 2023

In 2013, Baltic Earth was launched with a new science plan to strengthen efforts to address Grand Challenges on
(1) salinity dynamics in the Baltic Sea, (2) biogeochemical linkages between land and sea, (3) natural hazards and
extreme events, (4) sea level and coastal dynamics, (5) regional variability in water and energy exchanges, and (6)
multiple drivers of regional Earth system changes (Meier et al., 2014). Working groups on coupled Earth system
models, the Baltic Sea Model Intercomparison Project (BMIP), uncertainty of scenario simulations for the Baltic
Sea, and education, outreach and communication have been established.

Baltic Earth and its predecessor BALTEX have produced three comprehensive regional assessment reports since
2008. The first two (The BACC Author Team, 2008, and The BACC II Author Team, 2015) focused on climate
change and its impacts in the Baltic Sea region and were published as text books, while the third, the Baltic Earth
Assessment Reports (BEARs), was published in the format of a special issue of the journal *Earth System Dynamics*
in 2022. The Assessment of Climate Change in the Baltic Sea Basin (BACC) reports[3] and BEARs fill a gap
compared to the assessment reports of the Intergovernmental Panel on Climate Change (IPCC), as the latter focus
on global scales, and do not provide detailed local to regional information on the current state of knowledge on
climate change and its impacts in the Baltic Sea region. The BEARs provide a comprehensive and up-to-date
overview of the state-of-the-art research on the compartments of the Earth system in the Baltic Sea region
encompassing processes in the atmosphere, on land and in the sea, including the marine and terrestrial ecosystems
as well as processes and impacts related to the anthroposphere.

The BEARs summarise the published scientific knowledge currently available and update the second BACC report
(The BACC II Author Team, 2015) based on the latest scientific literature. This BEAR special issue includes 10
articles on the Baltic Earth Grand Challenges and Baltic Earth Special Topics (Baltic Earth Science Plan, 2017),
including a summary of current knowledge on past, present, and future climate change in the Baltic Sea region.
The articles encompass contributions from 109 authors from 14 countries and reference 2822 scientific articles
and institutional reports.
**1.2 Baltic Sea region characteristics**
The Baltic Sea is a semi-enclosed, shallow sea with limited water exchange with the World Ocean and small tidal
amplitudes. Located in Northern Europe, the climate of the region is highly variable as it is in the transition zone
between maritime and continental climates and is influenced by the North Atlantic and Arctic. River discharges
from the large catchment area cause a pronounced gradient in sea surface salinity from about 20 g kg$^{-1}$ in the
Danish straits' region to about 2 g kg$^{-1}$ or even less in the northern and eastern reaches of the Baltic Sea. Hence,
the Baltic Sea is brackish, with habitats of marine species in the south-west and freshwater species in the north-
east. The Baltic Sea catchment area is about four times the surface area of the Baltic Sea and covers an area of
almost 20% of the European continent (Fig. 1). It stretches from the temperate, densely populated south to the

---

[3] Assessment of Climate Change in the Baltic Sea Basin (BACC); https://baltic.earth/bacc, last access: 4 February 2023

subarctic wilderness in the north and is home to approximately 85 million people in 14 countries, namely Belarus,
the Czech Republic, Denmark, Estonia, Finland, Germany, Latvia, Lithuania, Norway, Poland, Russia, Slovakia,
Sweden and Ukraine.

Episodically, large amounts of saline water flow from the North Sea over the sills in the Danish straits into the
Baltic Sea and ventilate the deep waters of the Baltic Sea. These events require a period of about 20 days with
easterly winds that lower the sea level in the Baltic Sea, followed by a period of about the same length with strong
westerly winds that push saline water into the Baltic Sea. These events are called Major Baltic Inflows (MBIs) and
are important for the water exchange between the North Sea and the Baltic Sea. Mixing is low compared to other
seas, with an origin at the lateral boundaries, because tidal amplitudes are very small and energetically
insignificant.

In recent decades, environmental conditions in the Baltic Sea have changed considerably. For instance, the Baltic
Sea has been warming more than any other coastal sea since 1980 (Fig. 2), which has led to a reduction in sea ice
and snow cover over the land in winter. Furthermore, increasing nutrient input from the land in the 1950s/60s,
caused by population growth and the discharge of sewage into the Baltic Sea, as well as the increased use of
fertilisers in agriculture, led to eutrophication and the spread of hypoxic and anoxic areas. Since the 1980s, nutrient
inputs into the Baltic Sea have been steadily decreasing, but this has not yet led to a significant improvement in
oxygen conditions. Recent trends in acidification are lower than in the World Ocean, especially in the northern
Baltic Sea, as positive trends in alkalinity input counteract acidification.
**2 Methods**
Succeeding The BACC Author Team (2008) and The BACC II Author Team (2015) assessments, the BEAR
project is an attempt to summarise the scientific knowledge on climate change and other drivers of Earth system
changes and their impacts on the Baltic Sea region. The two BACC books have a format inspired by the IPCC
assessment reports. This special issue in *Earth System Dynamics* is the third assessment. It has a new format of
BEARs, encompassing 10 peer-reviewed scientific journal articles. The knowledge assessed was extracted from
the scientific literature such as peer-reviewed articles, reports from research institutions, and published datasets.
Importantly, literature from non-governmental organisations with political or economic interests, political parties
and other stakeholder organisations was excluded from the assessment to ensure that only scientific knowledge
was included in the assessment. The BEARs focus on publications after 2012/2013, the year of the editorial
deadline of the second assessment report. Whenever possible, the uncertainty levels of the BEAR results are ranked
based on a matrix of consensus within the scientific literature and documented evidence of detected changes and
their attributed drivers such as climate change and human use. A high level of scientific consensus and evidence
is required for high confidence in a particular statement. Disagreements and gaps in knowledge are documented
and discussed to prioritise future research.

Together with the intergovernmental Baltic Marine Environment Protection Commission (HELCOM), Baltic
Earth has established an Expert Network on Climate Change (EN CLIME). The aim of the expert network is to
regularly produce a climate change fact sheet (CCFS, 2021[4]) based on the BEAR and BACC material. In 2021, it
was published for the first time[5]. The CCFS contains some background information, a map showing regional future
climate changes for selected parameters under the greenhouse gas concentration scenario RCP4.5 and information
on 34 variables, directly and indirectly affected by climate change. For each parameter, a general description, past
and prospective future changes, other drivers than climate change (only for the indirect parameters), knowledge
gaps, policy relevance and references are presented. More than 100 scientists contributed to the compilation of the
first fact sheet, which was coordinated by the HELCOM secretariat. Updated versions are planned at seven-year
intervals. Like the BEARs, the fact sheet was peer-reviewed and quality assured. It has so far only been translated
to German (Klimawandel in der Ostsee, 2021 Faktenblatt, 2022[6]), but translations into other languages are planned
to improve accessibility to stakeholders.

In this editorial, we highlight the key findings and knowledge gaps as described by the BEARs and propose future
work.
**3 Results**
Some of the key findings of the 10 BEARs are selected and highlighted below.

1.  **Salinity dynamics of the Baltic Sea,** Grand Challenge 1 (Lehmann et al., 2022): Salinity is an important
parameter for the circulation and the marine ecosystem in the Baltic Sea. Any changes in salinity are
caused by changes either in the freshwater inflow from rivers and net precipitation over sea or in the water
exchange between the Baltic Sea and the adjacent North Sea. Although long-term records of salinity and
its drivers suffer from data gaps, these records starting in the 19th century are globally unique. Major
research efforts focused on the MBI event in 2014 and its consequences for water masses, oxygen
concentration and biogeochemical cycling. During the event, an unexpectedly large contribution of oxic
intrusions at intermediate depth and essentially nonturbulent conditions in the deep interior were found,
emphasising the importance of boundary mixing. A revised reconstruction of the long-term record of
MBIs showed no trend but a pronounced multi-decadal variability with a period of about 30 years. Despite
intense research activities, observed variations in the intensity and frequency of MBIs and related Large
Volume Changes (LVCs) could not be attributed to atmospheric circulation variability. Hence, on time
scales larger than the synoptical time scale, MBIs are not predictable. As an advance over the previous

[4] https://helcom.fi/wp-content/uploads/2021/09/Baltic-Sea-Climate-Change-Fact-Sheet-2021.pdf, last access: 4 February 2023

[5] http://helcom.fi/ccfs, last access: 4 February 2023

[6] https://baltic.earth/ccfs, last access: 4 February 2023

assessments, salinity dynamics of the various sub-basins and lagoons mainly based on observations have been discussed, documenting large regional differences.

2. **Biogeochemical functioning of the Baltic Sea**, Grand Challenge 2 (Kuliński et al., 2022): The review addresses the following topics: (1) terrestrial biogeochemical processes and nutrient inputs to the Baltic Sea, (2) the transformation of C, N and P in the coastal zone, (3) the production and remineralisation of organic matter, (4) oxygen availability, (5) the burial and turnover of C, N and P in sediments, (6) the Baltic Sea $CO_2$ system and seawater acidification, (7) the role of certain microorganisms in the biogeochemistry of the Baltic Sea, and (8) the interactions between biogeochemical processes and chemical pollutants. It was found that oxygen depletion and the area of anoxic bottoms have still increased despite the reductions in nutrient inputs from land since the 1980s. Hence, the nitrogen pool has declined due to denitrification whereas the phosphorus inventory has increased. Estimates suggest that about 1% and 4% of the annual nitrogen and phosphorus loads, respectively, have accumulated in the Baltic Sea, while the remainder are either exported to the North Sea or lost via biogeochemical processes such as denitrification and burial. Furthermore, it was discovered that in the central and northern sub-basins the uptake of C, N and P during primary production does not correspond to the Redfield ratio, which strongly affects the relationship between primary production, export of organic matter and oxygen demand of the deep sea. While it is clear that the Baltic Sea is a $CO_2$ sink in summer and a $CO_2$ source in winter, the annual net balance remains unknown. The past increase in total alkalinity of unknown origin has entirely mitigated ocean acidification in the northern Baltic Sea and significantly reduced it in the central Baltic Sea. In the future, a doubling of atmospheric $pCO_2$ would still result in lower pH in the entire Baltic Sea, even if alkalinity should further increase.

3. **Natural hazards and extreme events in the Baltic Sea region**, Grand Challenge 3 (Rutgersson et al., 2022): Existing knowledge is summarised about extreme events in the Baltic Sea region with a focus on the past 200 years with instrumental data as well as on future projections. Considered events are wind-storms, extreme waves, high and low sea levels, hot and cold spells in the atmosphere, marine heat waves, droughts, sea-effect snowfall, sea-ice ridging, extremely mild and extremely severe sea ice winters, heavy precipitation events, river floods, and extreme phytoplankton blooms. Furthermore, the knowledge about implications of these extreme events for society such as forest fires, coastal flooding, offshore infrastructure and shipping was assessed. With respect to the impacts of climate change, terrestrial and marine heat waves, extremely mild sea ice winters, heavy precipitation and high-flow events are expected to increase, while cold spells, severe sea ice winters and sea-ice ridging are expected to decrease due to the increase in mean atmospheric temperature. Changes in relative sea level extremes will depend on the competing impacts of the rising global mean sea level, the gravitational effect of the melting of the Greenland and Antarctic ice sheets, changes in wind fields, and the regionally differing Glacial Isostatic Adjustment (GIA) resulting in land uplift or subsidence. Furthermore, projections suggest an increase of droughts in the southern and central parts of the Baltic Sea region mainly in summer. Significant future

changes in wind-storms, extreme waves and sea level extremes relative to the mean sea level have not been found, suggesting that these changes will likely be small compared with natural variability.

4. **Sea level dynamics and coastal erosion in the Baltic Sea region**, Grand Challenge 4 (Weisse et al., 2021): In this study, the current knowledge about the diverse processes affecting mean and extreme sea level changes, coastal erosion and sedimentation with impact on coastline changes and coastal management is assessed. Such processes are GIA, contributions from global sea level changes, wind-storms, wind-waves, seiches or meteotsunamis. During 1886-2020, the mean absolute sea level in the Baltic Sea corrected for GIA increased by about 25 cm or ~2 mm year$^{-1}$ on average. Land uplift in the north is still faster than the absolute sea level rise while in the south the opposite is true with potential impacts on changes in coastal erosion and inundation. The current acceleration of sea level rise is small and could only be determined by spatially averaging observations at different tide gauge locations. Future sea level rise in the Baltic Sea is expected to further accelerate, probably somewhat less than the global mean sea level rise. The Baltic sea level is substantially more sensitive to melting from the Antarctic than from the Greenland ice sheet. Concerning sediment transports, the dominance of mobile sediments makes the southern and eastern Baltic Sea coasts susceptible to wind-wave induced transports, in particular during storms. Due to the global sea level rise, future sediment transports can be expected to increase in these coastal areas, with a large spatial variability depending on the angles of incidence of incoming wind-waves.

5. **Human impacts and their interactions in the Baltic Sea region**, Grand Challenge 6 (Reckermann et al., 2022): An inventory and discussion of the various man-made factors and processes affecting the environment of the Baltic Sea region and their interrelationships are presented. In total, 19 factors are addressed (Table 1). Some of the factors are natural and are only modified by human activities (e.g. climate change, coastal processes, hypoxia, acidification, submarine groundwater discharges, marine ecosystems, non-indigenous species, land use and land cover), others are entirely man-made (e.g. agriculture, aquaculture, fisheries, river regulation, offshore wind farms, shipping, chemical contamination, dumped ammunition, marine litter and microplastics, tourism and coastal management). All factors are interconnected to varying degrees. The knowledge of these linkages was assessed and analysed in depth. The main finding was that climate change has an overarching, integrating effect on all other factors and can be interpreted as a background effect that affects the other factors differently. After climate change, shipping and land use/agriculture are the factors affecting most other factors, while fisheries, marine ecosystems and agriculture in turn are the most affected. The results of the assessment depend on the region and may be different for other coastal seas and their catchments in the world, where different human activities prevail.

6. **Global climate change and the Baltic Sea ecosystem: direct and indirect effects on species, communities and ecosystem functioning**, Baltic Earth Special Topic (Viitasalo and Bonsdorff, 2022): Climate change has multiple impacts on species, communities and ecosystem functioning in the Baltic

Sea through changes in physical and biogeochemical parameters such as temperature, salinity, oxygen,
pH and nutrient levels. The associated secondary effects on species interactions, trophic dynamics and
ecosystem function are also likely to be important. Climate change (warming, recent brightening,
decrease in sea ice) has led to shifts in the seasonality of primary production, with a prolonged growing
season of phytoplankton, an earlier onset of the spring bloom and a delayed autumn bloom. However, the
development of cyanobacteria varies from species to species, and a clear causal relationship between
temperature or salinity and the abundance of cyanobacteria has not been demonstrated. An increase in
water temperature and river input of dissolved organic matter (DOM) could reduce primary production
while favouring bacterial growth. If nutrient reduction continues, the improvement in oxygen conditions
could initially increase zoobenthos biomass, but the subsequent decrease in sedimenting organic matter
would likely disrupt the pelagic-benthic coupling and result in lower zoobenthos biomass. Sprat and some
coastal fish species could be favoured by a rise in temperature. Regime shifts and cascading effects have
already been observed in both pelagic and benthic systems as a result of climate change.
7. **Coupled regional Earth system modeling in the Baltic Sea region**, Baltic Earth Special Topic, with
relevance to Baltic Earth Grand Challenge 5 (Gröger et al., 2021): Recent progress in the development of
coupled climate models for the Baltic Sea region is assessed. Feedback mechanisms are important to
simulate the response of the Earth system to external forcing such as greenhouse gas and aerosol
emissions. In this review article, the couplings between (1) atmosphere, sea ice and ocean, (2) atmosphere
and land surface including dynamic vegetation, (3) atmosphere, ocean and waves and (4) atmosphere and
hydrological components to close the water cycle are discussed. Adding surface waves to coupled
atmosphere-ocean system models is becoming more important with increasing resolution, in particular
when detailed information is required, for instance, for offshore wind energy applications in the coastal
zone. Furthermore, the wave information is essential for the calculation of ocean mixing and
resuspension. While long-term climate simulations using coupled atmosphere, sea ice and ocean models
or coupled atmosphere and dynamic vegetation models have successfully been performed and their added
value demonstrated, the impact of aerosols on the climate of the Baltic Sea region has not been considered.
Coupling hydrology models to close the hydrological cycle is also still problematic, as the precipitation
accuracy provided by the atmospheric models is, in most cases, insufficient to realistically simulate river
discharge into the Baltic Sea without bias adjustments.
8. **Atmospheric regional climate projections for the Baltic Sea region until 2100**, Baltic Earth Special
Topic (Christensen et al., 2022): Current climate projections based on regional climate atmosphere-only
models of the EURO-CORDEX project with a horizontal resolution of 12.5 km under the scenarios
RCP2.6, 4.5 and 8.5 are presented. As the number of simulations (124) is relatively large compared to
previous assessments, the uncertainties can be better estimated than before. These projections indicate
strong warming, especially in the north in winter, where warming approaches twice the average global
warming. Precipitation is projected to increase throughout the Baltic Sea region, except in the southern
half in summer, where the results are inconclusive. Extreme precipitation, here the 10-year return value,

is projected to increase systematically throughout the study area, especially in summer. The large ensemble of simulations does not indicate a significant change in wind speed. Surface solar radiation is projected to remain unchanged in summer, but to decrease slightly in winter, due to increased cloud cover and possibly less snow in the future. Snow cover is projected to decrease dramatically, especially in the south of the Baltic Sea catchment. The comparison between the uncoupled model simulations of the EURO-CORDEX project and a small ensemble of scenario simulations performed with a coupled atmosphere-sea-ice-ocean model driven by a subset of global climate models indicates stronger warming in the coupled model during winter, mainly in areas that are seasonally affected by sea ice today. In summer, the coupled model shows weaker warming compared to the uncoupled models.

9. **Oceanographic regional climate projections for the Baltic Sea until 2100**, Baltic Earth Special Topic (Meier et al., 2022a): New projections of the future Baltic Sea climate with a coupled physical-biogeochemical ocean model were compared with previous projections. The differences are mainly due to different scenario assumptions and model setups. For example, the impact of future global sea level rise on salinity was previously neglected, but taken into account in the latest projections. Although the number of projections for the Baltic Sea is still small compared to regional atmospheric projections such as the EURO-CORDEX model ensemble, a relatively large ensemble of 48 scenario simulations allowed the assessment of uncertainties related to greenhouse gas emissions, global climate model differences, global sea level rise, nutrient inputs and natural variability. In the future climate, higher water temperatures, a shallower mixed layer with a sharper thermocline in summer, lower sea ice cover and stronger mixing in the northern Baltic Sea in winter compared to the current climate could be expected. The assessment of marine heat wave changes is new. Both the frequency and duration of marine heat waves are projected to increase significantly, especially in the coastal zone of the southern Baltic Sea. Due to uncertainties in the projections regarding regional winds, precipitation and global sea level rise, no robust and statistically significant changes in salinity could be identified. The impacts of a changing climate on the biogeochemical cycle are projected to be significant, but still less than the plausible changes in nutrient inputs. Implementation of the proposed Baltic Sea Action Plan, a basin-wide nutrient input reduction plan, would lead to a significant improvement in the ecological status of the Baltic Sea, including a reduction in the size of the hypoxic area also in a future climate.

10. **Climate change in the Baltic Sea region: A summary**, Baltic Earth Special Topic (Meier et al., 2022b): In this comprehensive study, the recent knowledge on past (paleo-), present (historical) and projected future (< 2100) climate change in the Baltic Sea region, based upon all BEARs and >800 scientific articles, is summarised. It focuses on the atmosphere, the land surface, the cryosphere, the ocean and its sediments, and the terrestrial and marine biospheres. 33 parameters characterising the state of these components of the Earth system were analysed (Fig. 3, Table 2). The anthroposphere is not part of this assessment by Meier et al. (2022b) but instead is discussed in detail by Reckermann et al. (2022). The main findings concerning changes of the 33 selected state parameters attributed to climate change are summarised in Figure 3. The prevailing causal relationships of climate change with sufficiently high

confidence suggest a clear impact of global greenhouse gas emissions on regional heat cycles including all parameters of the cryosphere. However, changes caused by global warming of the water, momentum and carbon cycles are less clear because of either the large natural variability at regional scales or the impact of other drivers than global warming. For further details, the reader is referred to Meier et al. (2022b). Overall, it was concluded that the results from the previous BACC assessments mainly are still valid. However, new long-term, homogenous observational records, such as those for Scandinavian glacier inventories, sea-level-driven saltwater inflows (MBIs), or phytoplankton species distributions, and new scenario simulations with improved models, such as those for glaciers, lake ice, or marine food webs, have become available, resulting in a revised understanding of observed changes. Compared to previous assessments, observed changes in air temperature, sea ice, snow cover, and sea level were shown to have accelerated. However, natural variability is large, challenging our ability to detect observed and projected changes in climate of the Baltic Sea region. As the ensembles of scenario simulations both for the atmosphere and the ocean became larger, uncertainties can now be better estimated, although coordinated scenario simulations for the Baltic Sea based on ensembles of different regional ocean models are still missing. Furthermore, with the help of coupled models, feedbacks between several components of the Earth system have been studied, and multiple driver studies were performed, e.g., projections of the marine food web that include fisheries, eutrophication and climate change. Intensive research on the land–sea interface, focusing on the coastal filter, has been performed, and nutrient retention in the coastal zone was estimated for the first time. However, a model for the entire Baltic Sea coastal zone is still missing, and the effect of climate change on the coastal filter capacity is still unknown. More research on changing extremes was performed, acknowledging that the impact of changing extremes may be more important than that of changing means (see also Rutgersson et al., 2022). However, many observational records are either too short or too heterogeneous for statistical studies of extremes due to data gaps.

**4 Discussion**

One of the main objectives of the BEAR project was to identify knowledge gaps in the Earth system science of the Baltic Sea region so that these can be further addressed in future research. For specific knowledge gaps that have been identified during the project, the reader is referred to the individual assessment reports. However, as an overarching result, three new research topics are identified:

1) **Small-scale processes and their impact on large-scale climate dynamics and biogeochemical cycles.** The number of observations in the sea is smaller than those on land. This is also true for the Baltic Sea, although the international long-term monitoring programme in the Baltic Sea started more than a century ago, with measurements of temperature, salinity and oxygen concentration in the central parts of the different sub-basins. Nowadays, monitoring data are available from all sub-basins with a resolution of up to one month. Recently, many new observational systems for temporally and spatially high-resolution data have been developed or are under development, including remotely operated vehicles (ROVs) and autonomous underwater vehicles (AOVs) as well as remote sensing data. Examples of such systems

operating in the Baltic Sea are continuously profiling moorings, ARGO floats, Gliders, ScanFish, and echo sounders. In addition to traditional physical parameters, measurements of turbulence, biogeochemical and biodiversity (e.g. environmental DNA) parameters are now available. Another area of research that is developing rapidly is numerical modelling of the Earth system, also on a regional scale, e.g. eddy- and submesoscale resolving multi-year simulations for the Baltic Sea. Similar arguments apply to the atmosphere, e.g. cloud-resolving simulations to cope with heavy precipitation events. A novel research topic for Baltic Earth would therefore be a better understanding of the dynamics of small-scale atmospheric and oceanic processes that are not yet resolved in state-of-the-art numerical models or conventional observations, and their role in the large-scale circulation on short and long time scales. Such research activities would help to fill some of the gaps in knowledge that have been raised by Lehmann et al. (2022), Kuliński et al. (2022), Rutgersson et al. (2022), Weisse et al. (2021), Viitasalo and Bonsdorff (2022), and Gröger et al. (2021). Furthermore, a realistic consideration of small-scale processes would improve the projections for the atmosphere (Christensen et al., 2022) and the ocean (Meier et al., 2022a).

2) **Attribution of regional climate variability and change to anthropogenic radiative forcing and other drivers.** In order to unambigiously disentangle the impacts of anthropogenic climate change and other human influences from the natural climate variability of the regional Earth system, more knowledge about internal variations and feedback mechanisms is needed. For example, climate models have recently shown that multi-decadal variability emanating from the North Atlantic and the Arctic significantly controls the climate of the Baltic Sea region by means of teleconnection patterns (Lehmann et al., 2022; Meier et al., 2022a; 2022b). For example, observations of precipitation and wind in the Baltic Sea region, total river discharge from the catchment, individual river flows, water temperature, sea level, MBIs and salinity in the Baltic Sea show a pronounced multidecadal variability with a quasi-periodicity of about 30 years (Meier et al., 2022b). It is assumed that the Atlantic Multidecadal Variability (AMV) and, as part of it, the variations of the North Atlantic overturning circulation are the source of these variations, although the exact mechanisms, cause and effect chains and feedback processes are still unknown. Knowledge about the teleconnectivity of the Baltic Sea region with the North Atlantic and the Arctic is essential for the development of climate prediction models.

3) **Development of integrated Earth system models accounting for anthropogenic changes in the Baltic Sea region.** The BEAR study by Reckermann et al. (2022) on human influences and their interactions in the Baltic Sea region is part of the relatively new Baltic Earth Grand Challenge 6 on the multiple drivers of Earth system change in the Baltic Sea region and represents an important step towards an integrated understanding of the Earth system that encompasses all traditionally considered climate compartments such as atmosphere, cryosphere, hydrosphere, lithosphere (including the pedosphere), biosphere (marine and terrestrial) and the anthroposphere. Such a holistic view is urgently needed, as in many cases, several reasons are responsible for the observed changes in the Earth system and attributing them to only one factor, e.g. climate change would be an inadmissible simplification. One example is the oxygen depletion and the large hypoxic area in the Baltic Sea caused by anthropogenic nutrient inputs from land and

exacerbated by rising water temperatures (Kuliński et al., 2022). Of course, the factors discussed by
Reckermann et al. (2022) cannot exhaustively consider the entire Earth system and all interactions, and
the selection of factors is biased towards ocean-related parameters and activities. Moreover, the analysis
is based on an extensive literature review by experts who reflect their subjective interpretations of the
results. Nevertheless, this is the first time such an assessment has been conducted, which is a major step
forward. To continue and deepen this research, the factors discussed by Reckermann et al. (2022) could
be subdivided either by human activities (e.g. food production, energy production, transport, tourism,
healthcare) or by environmental and climate state variables of the Earth system (e.g. hypoxia,
acidification) (Table 3). Such a breakdown of parameters would allow the development of an integrated
Earth system model that includes the anthroposphere at the regional scale. This type of research is timely,
and such efforts are already underway (e.g. Korpinen et al., 2019; references in Reckermann et al., 2022).

The fact sheet on climate change in the Baltic Sea (CCFS, 2021) was positively received by various stakeholders
and decision-makers. Although uncertainties regarding observed and projected future climate change and the other
drivers remain high, our experience engaging with stakeholders confirms that scientific uncertainties are taken into
account in different ways in management and decision-making. This is an important reason for investing in the
above key issues. They have the potential to reduce uncertainties that currently hamper decision-making in the
region.
**5 Concluding remarks**
We conclude that 1) the BEARs have been useful to identify research progress and knowledge gaps and to initiate
new research foci as, for examples, suggested in the discussions; 2) regional assessments, such as the BEARs,
complement the IPCC climate change assessments by adding a greater depth and scope of regional information
about the specific situation of the Baltic Sea region; and 3) the BEARs provided useful information for the Expert
Network on Climate Change, that produced the Baltic Earth – HELCOM climate change fact sheet for
stakeholders. Since the information summarised by the BEARs are used extensively in science and management,
it is recommended that a new update of the reports will be conducted in about seven years.
**Author contributions**
H.E.M.M. wrote the first draft of the editorial. All co-authors, which acted as guest editors of the special issue in
*Earth System Dynamics*, contributed with important comments and editing of the manuscript, read and approved
the submitted manuscript version.
**Acknowledgements**
During 2019-2022, the Baltic Earth Assessment Reports were produced under the umbrella of the Baltic
Earth programme (Earth System Science for the Baltic Sea region, see http://baltic.earth, last access: 2
February 2023). 109 co-authors from 14 countries contributed to 10 articles in the international scientific

journal *Earth System Dynamics* and 2822 different references have been assessed. We thank the reviewers of all 10 articles of the special issue for their constructive comments that helped to improve the review articles. In particular, we thank Dr. Jouni Räisänen, Dr. Donald Boesch and Dr. Andris Andrusaitis for their advice and many excellent comments on individual articles and this overview article.

**Figures**

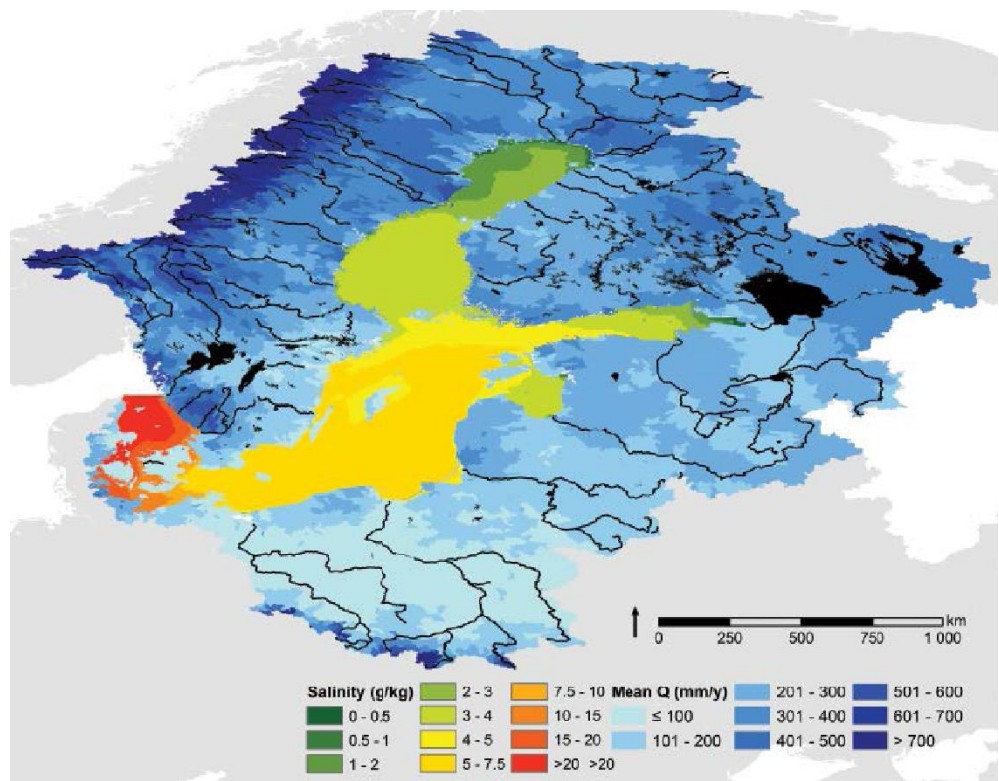


**Figure 1:** The Baltic Sea and its catchment area with climatological mean sea surface salinity (in g kg⁻¹) and river
discharge (in mm year⁻¹). (Source: Meier et al., 2014; their Fig. 1 distributed under the terms of the Creative
Commons CC-BY 4.0 License, http://creativecommons.org/licenses/by/4.0/, last access: 4 February 2023)


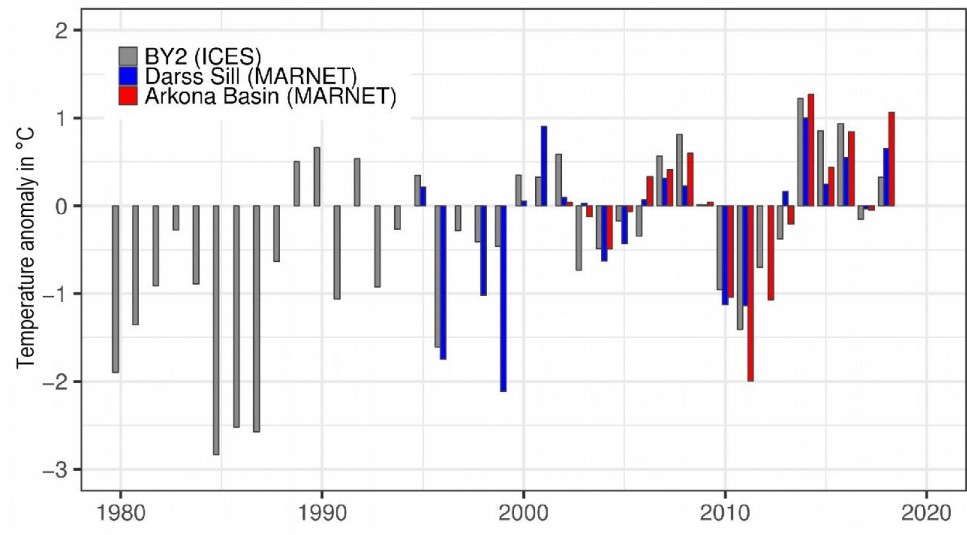


**Figure 2:** Annual mean sea surface temperature anomalies relative to the reference period 2002-2018 from de-
seasonalised measurements at the Arkona Deep monitoring station and the MARNET stations Darss Sill and
Arkona Basin in the period 1980-2018. (Source: Meier et al., 2022b; their Fig. 20 distributed under the terms of
the Creative Commons CC-BY 4.0 License, http://creativecommons.org/licenses/by/4.0/, last access: 4 February
430 2023)



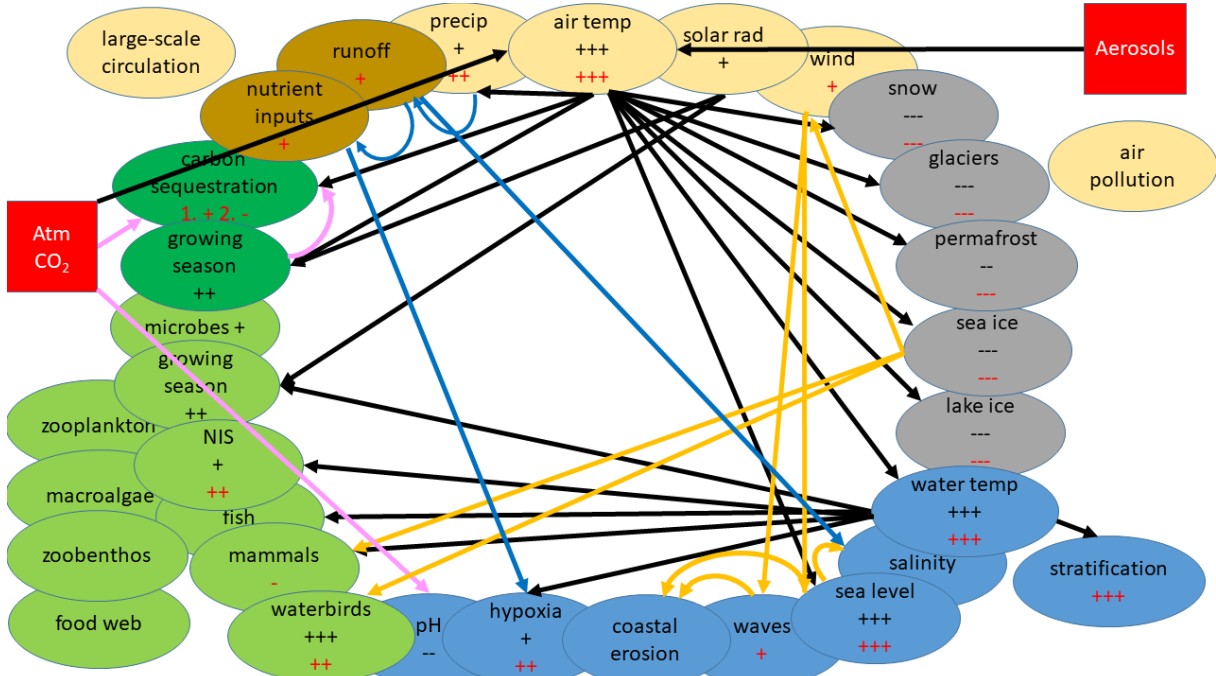


**Figure 3:** Synthesis of knowledge on present and future climate change. Shown are anthropogenic climate changes in 33 Earth system variables (bubbles) of the atmosphere (yellow), land surface (brown), terrestrial biosphere (dark green), cryosphere (grey), ocean and sediments (blue) and marine biosphere (light green). The abbreviation NIS stands for non-indigenous species. The sign of a change (plus/minus) is shown together with the confidence level indicated by the number of signs, i.e. one to three signs correspond to a low, medium and high confidence level as a result of the literature assessment reflecting consensus and evidence according to IPCC definitions. The colours of the signs indicate the direction of past (black) and future (red) changes according to Meier et al. (2022b). Uncertain changes are not shown. The external anthropogenic drivers of the Earth system studied are shown as red squares, i.e. greenhouse gases, especially $CO_2$, and aerosol emissions. The predominant climate change linkages with sufficiently high confidence are shown by arrows (black: thermal cycle, blue: hydrological cycle, orange: momentum cycle including sea level change, pink: carbon cycle). Projections of carbon sequestration of Arctic terrestrial ecosystems for the 21st century show first an increased uptake and later a carbon source, marked by "1. + 2. –". Future changes in mean sea level are dominated by the thermal expansion of the global ocean and the melting of ice sheets outside the Baltic Sea region. (Source: Meier et al., 2022b; their Fig. 35 distributed under the terms of the Creative Commons CC-BY 4.0 License, http://creativecommons.org/licenses/by/4.0/, last access: 4 February 2023)

450

**Table 1:** The matrix of factors studied by Reckermann et al. (2022). + = evidence for a connection; - = no evidence for a connection; ? = no evidence, but connection plausible (according to the author's assessment). The table is read from left to right, i.e. if you go to the right in the first row "climate change", you see the factors on which climate change has an effect (or not), etc. (Source: Reckermann et al., 2022; their Table 2a distributed under the terms of the Creative Commons CC-BY 4.0 License, http://creativecommons.org/licenses/by/4.0/, last access: 4 February 2023)

| impact by↓/on→ | Climate change | Coastal processes | Hypoxia | Acidification | Subm. Groundw. Disch. | Marine ecosystems | Non-inig. species | Land cover and use | Agriculture Nutr. loads | Aquaculture | Fisheries | River regulations | Offshore wind farms | Shipping | Chem. Contamin. | Dumped millitary | Marine litter | Tourism | Coastal management |
|---|---|---|---|---|---|---|---|---|---|---|---|---|---|---|---|---|---|---|---|
| Climate change | | + | + | + | ? | + | ? | + | + | + | + | + | + | + | + | ? | ? | + | + |
| Coastal processes | - | | ? | ? | + | ? | - | + | + | ? | ? | + | + | + | ? | ? | + | - | + |
| Hypoxia | - | - | | + | - | + | - | - | + | ? | + | - | - | - | + | + | - | - | - |
| Acidification | - | - | - | | - | ? | - | - | - | ? | ? | - | - | - | ? | - | - | - | - |
| Subm. Groundw. Disch. | - | - | ? | ? | | ? | - | - | + | - | - | - | - | - | + | - | - | - | - |
| Marine ecosystems | - | - | + | + | - | | + | - | - | - | + | - | - | - | - | - | - | + | - |
| Non-inigenous species | - | - | - | - | - | + | | - | - | - | + | - | - | + | + | - | - | - | ? |
| Land cover and use | + | - | + | + | + | ? | - | | + | - | + | + | ? | - | + | - | - | + | ? |
| Agriculture/Nutrient loads | + | - | + | + | + | + | - | + | | + | + | + | - | - | + | - | - | - | ? |
| Aquaculture | - | - | + | - | - | + | + | + | + | | ? | - | + | - | ? | - | ? | ? | + |
| Fisheries | - | - | ? | - | - | + | ? | - | ? | ? | | - | + | ? | - | - | + | ? | + |
| River regulations | - | + | ? | + | ? | + | - | ? | ? | ? | + | | - | - | ? | - | ? | - | + |
| Offshore wind farms | + | + | - | - | - | + | - | ? | ? | + | + | - | | + | ? | ? | ? | + | + |
| Shipping | + | + | - | + | - | + | + | - | + | ? | + | - | ? | | + | - | + | + | + |
| Chemical contaminants | - | - | - | - | - | + | - | - | + | + | + | - | - | - | | - | - | - | - |
| Dumped millitary material | - | - | - | - | - | ? | - | - | - | - | + | - | - | + | - | | - | - | ? |
| Marine litter | - | - | - | - | - | ? | - | - | - | ? | + | - | - | - | ? | - | | + | ? |
| Tourism | + | - | - | - | - | ? | - | + | + | - | - | - | + | + | - | - | + | | + |
| Coastal management | - | + | - | - | ? | ? | ? | ? | ? | ? | + | ? | + | + | - | + | ? | + | |

**Table 2:** Variables of the Meier et al. (2022b) assessment and further references to the BEARs (1: Lehmann et al., 2022; 2: Kuliński et al., 2022; 3: Rutgersson et al, 2022; 4: Weisse et al., 2021; 5: Reckermann et al, 2022; 6: Gröger et al, 2021; 7: Christensen et al, 2022; 8: Meier et al, 2022a; 9: Viitasalo and Bonsdorff, 2022). The third column lists the subsection in the study by Meier et al. (2022b) that contains further information. (Source: Meier et al., 2022b; their Table 2 distributed under the terms of the Creative Commons CC-BY 4.0 License, http://creativecommons.org/licenses/by/4.0/, last access: 4 February 2023 )

| Number | Variable | Past and present climates | | Future climate | |
|---|---|---|---|---|---|
| Atmosphere | | | | | |
| 1 | Large-scale atmospheric circulation | 3.2.1.1 | 3 | 3.3.1.1 | 3, 7 |
| 2 | Air temperature | 3.1.2, 3.1.3, 3.1.4 | | 3.3.1.2 | 7 |
| | Warm spell | 3.2.1.2 | 3 | | 3 |
| | Cold spell | | 3 | | 3 |
| 3 | Solar radiation and cloudiness | 3.2.1.3 | | 3.3.1.3 | 7 |
| 4 | Precipitation | 3.1.2, 3.1.3, 3.1.4 | | 3.3.1.4 | 7 |
| | Heavy precipitation | 3.2.1.4 | 3 | | 3 |
| | Drought | | 3 | | 3 |
| 5 | Wind | 3.2.1.5 | | 3.3.1.5 | 7 |
| | Storm | | 3 | | 3 |
| 6 | Air pollution, air quality and atmospheric deposition | 3.2.1.6 | | 3.3.1.6 | |
| Land | | | | | |
| 7 | River discharge | 3.2.2.1 | | 3.3.2.1 | 8 |
| | High flow | | 3 | | 3 |
| 8 | Land nutrient inputs | 3.2.2.2 | | 3.3.2.2 | 8 |
| Terrestrial biosphere | | | | | |
| 9 | Land cover (forest, crops, grassland, peatland, mires) | 3.2.3 | 6 | 3.3.3 | |
| 10 | Carbon sequestration | | | 3.3.3 | |
| Cryosphere | | | | | |

| 11 | Snow Sea-effect snowfall | 3.2.4.1 | 3 | 3.3.4.1 | 7 3 |
|---|---|---|---|---|---|
| 12 | Glaciers | 3.2.4.2 | | 3.3.4.2 | |
| 13 | Permafrost | 3.2.4.3 | | 3.3.4.3 | |
| 14 | Sea ice Extreme mild winter Severe winter Ice ridging | 3.2.4.4 | 3 3 3 | 3.3.4.4 | 8 3 3 3 |
| 15 | Lake ice | 3.2.4.5 | | 3.3.4.5 | |
| Ocean and marine sediments | | | | | |
| 16 | Water temperature Marine heat wave | 3.2.5.1 | 3 | 3.3.5.1 | 8 3 |
| 17 | Salinity and saltwater inflows | 3.2.5.2 | 1 | 3.3.5.2 | 8 |
| 18 | Stratification and overturning circulation | 3.2.5.3 | 1 | 3.3.5.3 | 8 |
| 19 | Sea level Sea level extreme | 3.2.5.4 | 4 3 | 3.3.5.4 | 8 3 |
| 20 | Waves Extreme waves | 3.2.5.5 | 4 3 | 3.3.5.5 | 3 |
| 21 | Sedimentation and coastal erosion | 3.2.5.6 | 4 | 3.3.5.6 | |
| 22 | Oxygen and nutrients | 3.1.4 3.2.5.7.1 | 2 | 3.3.5.7.1 | 8 |
| 23 | Marine $CO_2$ system | 3.2.5.7.2 | 2 | 3.3.5.7.2 | |
| Marine biosphere | | | | | |
| 24 | Pelagic habitats: Microbial communities | 3.2.6.1.1 | 2, 9 | 3.3.6.1.1 | 9 |
| 25 | Pelagic habitats: Phytoplankton and cyanobacteria | 3.2.6.1.2 | 2, 3, 9 | 3.3.6.1.2 | 3, 9 |
| 26 | Pelagic habitats: Zooplankton | 3.2.6.1.3 | 9 | 3.3.6.1.3 | 9 |
| 27 | Benthic habitats: Macroalgae and vascular plants | 3.2.6.2.1 | 9 | 3.3.6.2.1 | 9 |

| 28 | Benthic habitats: Zoobenthos | 3.2.6.2.2 | 9 | 3.3.6.2.2 | 9 |
|----|------------------------------|-----------|---|-----------|---|
| 29 | Non-indigenous species | 3.2.6.3 | 9 | 3.3.6.3 | 9 |
| 30 | Fish | 3.2.6.4 | 9 | 3.3.6.4 | 9 |
| 31 | Marine mammals | 3.2.6.5 | 9 | 3.3.6.5 | 9 |
| 32 | Waterbirds | 3.2.6.6 | 9 | 3.3.6.6 | 9 |
| 33 | Marine food web | 3.2.6.7 | 9 | 3.3.6.7 | 9 |


**Table 3**: Factors discussed by Reckermann et al. (2022) sorted by related economic sectors or state variables of
the Earth system.

| Human activities | | |
|---|---|---|
| Economic sectors | Factors | Comments |
| Primary (natural ) sector (e.g. food production) | Fisheries | |
| | Agriculture | |
| | Marine and coastal ecosystem services | Factor belongs to several sectors |
| | Blue carbon storage capacity | Mitigation of greenhouse gases |
| Secondary (industrial) sector (e.g. energy production) | River regulation | |
| | Offshore wind farms | |
| | Greenhouse gas and aerosol emissions | Emission are largest from industries |
| | Dumped warfare agents | Factor is an industrial product |
| Tertiary (service) sector (e.g. transportation, tourism, healthcare) | Shipping | |
| | Chemical contamination | Contamination is a result of several sectors |
| | Marine noise | Marine noise is a result of several sectors |
| | Marine litter and microplastics | Emission mainly by offshore platforms, shipping, lost containers, fisheries, aquaculture, agriculture, municipal waste and tourism |
| | Tourism | |
| | Coastal protection and management | Also relevant for the other sectors |
| Quaternary (information) sector (e.g. information technology; media; research and development) | - | |
| Earth system | | |
| Environmental state variables | Coastal processes | |
| | Hypoxia | |
| | Submarine groundwater discharge | |
| | Marine ecosystems | |
| | Land use and land cover | |
| | Non-indigenious species | |

| | Indirect parameters such as carbon and nutrient cycles, biota and ecosystems | |
|---|---|---|
| Climate state variables | Climate change, acidification, direct parameters of the climate system | Superordinated concept (large-scale) |
| | Direct parameters of the climate system | |
| | Acidification | |

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
