# Peer review of "The Baltic Earth Assessment Reports 1"

_Earth System Dynamics, 2023_

## Author Comment (AC1)

**1. General comments**

This well-written and informative editorial summarises the main findings from the recently published Baltic Earth Assessment Reports (BEARs) and the most important knowledge gaps identified in them. It also nicely puts the BEARs in context, explaining their position in the history of Earth system research and climate change assessments for the Baltic Sea basin, as well as their connection to the stakeholder-oriented fact sheet that the Baltic Earth Expert Network on Climate Change has recently produced.

Overall, this is work well done. I only have a few minor comments on its substance and a few more suggestions for improving the wording and presentation.

Thank you very much for your positive and helpful comments. We have implemented all requested changes. Please find our detailed response in red below.

**2. Comments on substance and presentation**

1. What does "today successfully completed" refer to? Do you mean that all the work of the working groups has been completed or that scenario simulations have been completed?

The working group on scenario simulations has been completed. However, this is an unimportant detail. Hence, we have deleted the phrase in the brackets.

2. has been warming. "is warming" implicitly predicts that the observed faster warming can be extrapolated to the future, which may or may not be true.

Corrected.

3. L103-104. This sentence could be interpreted to mean that this is the third BEAR assessment. Suggested reformulation, beginning from L102: The two BACC books adopted a format inspired by the IPCC assessment reports. The present special issue in Earth System Dynamics is the third assessment. It takes a new format of Baltic Earth Assessment Reports (BEARs), encompassing 10 peer-reviewed scientific journal articles.

Thank you. We have adopted your suggestion.

4. I only found 30 years (but not 25) from Lehmann et al. (2022).

Corrected.

5. L147-148. As an advance over the previous assessments?

Very good suggestion, which we have implemented.

6. would still result (doubling of pCO2 is something that international climate policy very much strives to avoid).

Corrected.

7. L181-183. I think this sentence simplifies a bit too much. Even without GIA, the mean sea level rise in the Baltic Sea would differ from the global mean, due to (e.g.) the gravitational effect of the melting of Greenland ice. Furthermore, the changes in extremes might differ from those in the mean sea level, although (as noted on L185-186) there is no consensus that there would be a large difference.

We revised the sentence "Changes in relative sea level extremes will depend on the competition between the rising global mean sea level, the gravitational effect of the melting of the Greenland and Antarctic ice sheets, changes in wind fields, etc. and the regionally differing Glacial Isostatic Adjustment (GIA) resulting in land uplift or subsidence." The relative importance of the various drivers of changes in sea level extremes is not discussed here.

8. L186, If the ensemble size (rather than model weaknesses) is indeed the limiting factor, then there is no reason to expect that statistically significant changes would occur in the real world, where only one realization of climate change will be observed. Therefore, I would replace "probably because available climate model ensembles are too small" by something like "suggesting that these changes will likely be small compared with natural variability".

We agree and modified the sentence according to the reviewer's suggestion.

9. "Antarctic" should be "Greenland". Due to the gravitational effects, melting of Greenland will increase sea level much less in the Baltic Sea than in the global mean.

In the Baltic Sea, the fingerprint from the melting Antarctic ice sheet in the Baltic Sea is larger than from the Greenland ice sheet (Grinstedt, 2015, BACC II; his Fig. 14.2) because the zero line of the Greenland fingerprint pattern is located in the Baltic Sea. To avoid confusion, we deleted the sub-clause.

10. I guess this increase mainly affects the southern parts, and less if at all the north, where the absolute sea level rise and the GIA largely compensate each other. "increase in these coastal areas" would implicitly convey this.

Correct. We added "these"

11. Perhaps "recent brightening"? While brightening (i.e., increased solar radiation) has been observed in the last few decades, it was preceded by a general dimming between the Second World War and the 1980s, presumably due to increased aerosols.

We agree and added "recent"

12. L244-245. Adding surface waves to coupled atmosphere-ocean system models?

Corrected.

13. has not been considered?

Corrected.

14. L266-269. The lack of increasing solar radiation in summer is also most likely affected by the lack of time-varying aerosols in many of the RCMs. By contrast, the projected decrease in anthropogenic aerosol emissions in the GCM simulations leads to a future increase in solar irradiation (Boé et al. 2020, Climate Dynamics,

https://link.springer.com/article/10.1007/s00382-020-05153-1). If the projected decreases in aerosol emissions are realized, the GCM simulations should probably be judged more plausible regarding the changes in solar irradiation and perhaps also summer temperature.

We agree and deleted the misleading sentence. In Meier et al. (2022, Climate change in the Baltic Sea region: a summary) an explanation for the discrepancy is provided, following your argumentation.

15. cannot exhaustively take into account?

Corrected.

- 3. Technical comments
  - 1. Delete "more than". 19 is the exact number according to Table 1.

Corrected.

2. Biogeochemical

Corrected.

3. mean surface salinity?

Yes. Corrected.

- Figure 2. The y axis label should be "Temperature anomaly" rather than "Temperature".
   Corrected.
- 5. 2022a. Furthermore, the order of this and the previous reference (Meier et al., 2022b) should be switched.

Corrected.

Please find the revised manuscript below.

Thank you very much!

Markus on behalf of the co-authors

**The Baltic Earth Assessment Reports**

H. E. Markus Meier1, Marcus Reckermann2, Joakim Langner3, Ben Smith4,5 and Ira

Didenkulova6

1Department of Physical Oceanography and Instrumentation, Leibniz Institute for Baltic Sea Research Warnemünde, Rostock, Germany

2International Baltic Earth Secretariat, Helmholtz-Zentrum Hereon, Geesthacht, Germany

3Swedish Meteorological and Hydrological Institute, Norrköping, Sweden

4Hawkesbury Institute for the Environment, Western Sydney University, Australia

5Department of Physical Geography and Ecosystem Science, Lund University, Sweden

6Department of Mathematics, University of Oslo, Norway

Correspondence to: H. E. Markus Meier (markus.meier@io-warnemuende.de)

Abstract. Baltic Earth is an independent research network of scientists from all Baltic Sea countries that promotes regional Earth system research. Within the framework of this network, the Baltic Earth Assessment Reports (BEARs) were produced in the period 2019-2022. These are a collection of 10 review articles summarising current knowledge on the environmental and climatic state of the Earth system in the Baltic Sea region and its changes in the past (palaeoclimate), present (historical period with instrumental observations) and prospective future (until 2100) caused by natural variability, climate change and other human activities. The division of topics between articles follows the grand challenges and selected themes of the Baltic Earth Science Plan, such as the regional water, biogeochemical and carbon cycles, extremes and natural hazards, sea level dynamics and coastal erosion, marine ecosystems, coupled Earth system models, scenario simulations for the regional atmosphere and the Baltic Sea, and climate change and impacts of human use. Each review article contains an introduction, the current state of knowledge, knowledge gaps, conclusions and key statements, based on which recommendations are made for future research. In parallel, Baltic Earth's ongoing outreach work has led to the publication of an information leaflet on climate change in the Baltic Sea, which has been published in two languages so far, and the organisation of stakeholder conferences and workshops.

**1** Introduction**

**1.1 BALTEX/Baltic Earth history**

Baltic Earth is an international research network dealing with Earth system science of the Baltic Sea region (https://baltic.earth, last access: 4 February 2023). The catchment area of the Baltic Sea is about four times larger than the Baltic Sea surface and is part of mainly the countries Belarus, Denmark, Estonia, Finland, Germany, Latvia, Lithuania, Poland, Russia, and Sweden (Fig. 1). Baltic Earth is politically independent and focusses on research on the water and energy cycles, climate variability and climate change, water management and extreme events, and related impacts on marine and terrestrial biogeochemical cycles. Human impact on the Earth system in more general terms, i.e. the anthroposphere, defined as the part of the environment that is made or modified by humans for use in human activities, was added to the 2017 Baltic Earth Science Plan (https://baltic.earth/grandchallenges, last access: 4 February 2023).

Baltic Earth is the successor of the Baltic Sea Experiment (BALTEX) programme, which was founded in 1993 as a GEWEX continental-scale experiment (Global Energy and Water Exchanges, a core project of the World Climate Research Programme) (Reckermann et al., 2011). During Phase I (1993–2002), BALTEX was primarily devoted to hydrological, meteorological and oceanographic processes in the Baltic Sea drainage basin, hence focussed on physical aspects of the Earth system. During the second phase (Phase II: 2003–2012), the programme was expanded to encompass regional climate research, carbon and biogeochemical cycles, engagement with stakeholders and decision makers via assessment reports, as well as communication and education, i.e. organizing summer and winter schools and international master courses.

In 2013, Baltic Earth was launched with a novel science plan reinforcing efforts to address Grand Challenges on (1) salinity dynamics in the Baltic Sea, (2) land-sea biogeochemical linkages, (3) natural hazards and extreme events, (4) sea level and coastal dynamics, (5) regional variability of water and energy exchanges, and (6) multiple

drivers of regional Earth system changes (Meier et al., 2014). Working groups were initiated on coupled Earth system models, the Baltic Sea Model Intercomparison Project (BMIP), education, outreach and communication, and scenario simulations for the Baltic Sea.

Baltic Earth and its predecessor BALTEX have produced three extensive regional assessment reports since 2008. The first two (The BACC Author Team, 2008, and The BACC II Author Team, 2015) emphasised climate change and its impacts in the Baltic Sea region and were published as text books, while the third, the Baltic Earth Assessment Reports (BEARs), was published in the format of a special issue in *Earth System Dynamics*, in 2022. The BEARs and BACC assessment reports fill a gap relative to the assessment reports of the Intergovernmental Panel on Climate Change (IPCC), since the latter focus on global scales, and do not provide detailed local to regional information about the current state of knowledge on climate change and its impacts in the Baltic Sea region. The BEARs provide a comprehensive and up-to-date overview of the state-of-the-art research on the compartments of the Earth system in the Baltic Sea region encompassing processes in the atmosphere, on land and in the sea, including the marine and terrestrial ecosystems as well as processes and impacts related to the anthroposphere.

The BEARs wrap together the currently available published scientific knowledge, updating the second assessment report of climate change in the Baltic Sea basin (The BACC II Author Team, 2015) based on the scientific literature. The present BEAR Special Issue comprises 10 articles on the Baltic Earth Grand Challenges and working group topics including a summary of the current knowledge about past, present, and future climate changes for the Baltic Sea region. The articles encompass contributions by 109 authors from 14 countries and reference 2822 scientific articles and institutional reports in their synthesis effort.

**1.2 Baltic Sea region characteristics**

The Baltic Sea is a semi-enclosed, shallow sea with limited water exchange with the world ocean and small tidal amplitudes. Situated in Northern Europe, the climate of the region is highly variable because it is located in the transition zone between maritime and continental climates, influenced by the North Atlantic and Arctic regions. The river discharge from the large catchment area causes a pronounced gradient in sea surface salinity from about 20 g kg-1 in the Danish Straits region to about 2 g kg-1 or even less in the northern and eastern reaches of the Baltic Sea. Hence, the Baltic Sea is brackish, with habitats of maritime species in the south-west and freshwater species in the north-east. The Baltic Sea catchment area covers an area of almost 20% of the European continent. 85 million people in 14 countries live in the catchment area, which stretches from the temperate, densely populated south to the subarctic wilderness in the north.

Episodically, large amounts of saltwater from the North Sea cross the sills, located in the Danish Straits, into the Baltic Sea and ventilate the Baltic Sea deepwater. These events require a period of about 20 days of easterly winds that lower the Baltic sea level, followed by a period of about equal duration of strong westerly winds that push saltwater into the Baltic Sea. These events are called Major Baltic Inflows (MBIs) and are important for the water exchange between the North Sea and the Baltic Sea. Mixing is low compared to other seas, with an origin at the lateral boundaries, because tidal amplitudes are very small and energetically unimportant.

In recent decades, environmental conditions in the Baltic Sea have changed considerably. For instance, since 1980 the Baltic Sea has been warming more than any other coastal sea (Fig. 2), causing shorter sea ice and snow covers over land during winter. Furthermore, rising nutrient inputs from land in the 1950s/60s, caused by population growth accompanied by sewage water release to the Baltic Sea and intensified usage of fertilisers from agriculture, led to eutrophication and spreading of hypoxic and anoxic areas. Since the 1980s, nutrient input into the Baltic Sea has been steadily decreasing, but this has not yet led to a significant improvement in oxygen conditions. Recent trends in acidification are smaller than in the world ocean, in particular in the northern Baltic Sea, because positive trends in alkalinity supply counteract the acidification.

**2 Methods**

Following the BACC1 Author Team (2008) and the BACC II Author Team (2015), the BEAR project is an effort to summarise scientific knowledge about climate change and other drivers of Earth system changes and their impacts on the Baltic Sea region. The two BACC books adopted a format inspired by the IPCC assessment reports. The present special issue in Earth System Dynamics is the third assessment. It takes a new format of Baltic Earth Assessment Reports (BEARs), encompassing 10 peer-reviewed scientific journal articles. The assessed knowledge was extracted from the scientific literature such as peer-reviewed articles, reports from research institutions, and published datasets. Importantly, literature from non-governmental organisations with a political or economic interest, political parties and other stakeholder organisations was excluded from the assessment, ensuring that only scientific knowledge informed the assessment. The BEARs focus on publications after 2012/2013, the year of the editorial deadline of the second assessment report. Whenever possible, uncertainty levels of the BEAR results are classified based on a matrix of consensus within the scientific literature and the documented evidence of detected changes and their attributed drivers such as climate change and human use. For a high confidence of a certain statement, high levels of both scientific consensus and evidence cases are required. Instances of disagreement and knowledge gaps are documented and discussed, informing priorities for future research.

Together with the intergovernmental Baltic Marine Environment Protection Commission (HELCOM), Baltic Earth formed an Expert Network on Climate Change (EN CLIME). The aim of the expert network is to regularly produce a climate change fact sheet (CCFS, 20212) from the BEAR material. In 2021, it was published for the first time (http://helcom.fi/ccfs, last access: 4 February 2023). The CCFS contains some background information, a map showing regional future climate changes for selected parameters under the greenhouse gas concentration scenario RCP4.5 and information about 34 variables, directly and indirectly affected by climate change. For each parameter, a general description, past and prospective future changes, other drivers than climate change (only for the indirect parameters), knowledge gaps, policy relevance and references are presented. More than 100 scientists contributed to the compilation of the fact sheet which was coordinated by the HELCOM secretariat. Updated versions are planned for intervals of seven years. Like the BEARs, the fact sheet was peer-reviewed and quality assured. It has

<sup>1 Assessment of Climate Change in the Baltic Sea Basin (BACC); https://baltic.earth/bacc, last access: 4 February 2023

<sup>2 https://helcom.fi/wp-content/uploads/2021/09/Baltic-Sea-Climate-Change-Fact-Sheet-2021.pdf, last access: 4 February 2023

so far been translated to German (Ostsee Klimawandel Faktenblatt, 20223) and translations to other languages are planned, enhancing accessibility to stakeholders.

In this editorial, we highlight the main findings and knowledge gaps as detailed by the BEARs and future work is proposed.

**3 Results**

[revised manuscript text omitted]

**Acknowledgements**

During 2019-2022, the Baltic Earth Assessment Reports were produced under the umbrella of the Baltic Earth programme (Earth System Science for the Baltic Sea region, see <a href="http://baltic.earth">http://baltic.earth</a>, last access: 2 February 2023). 109 co-authors from 14 countries contributed to 10 articles in the international scientific journal Earth System Dynamics and 2822 different references have been assessed. We thank the reviewers of all 10 articles of the special issue for their constructive comments that helped to improve the review articles. In particular, we thank Dr. Jouni Räisänen and Dr. Donald Boesch for their advice and many excellent comments on the individual articles and this overview article.

**Figures**

**Figure 1:** The Baltic Sea and its catchment area with climatological mean sea surface salinity (in g kg-1) and river discharge (in mm year-1). (Source: Meier et al., 2014; their Fig. 1 distributed under the terms of the Creative Commons CC-BY 4.0 License, http://creativecommons.org/licenses/by/4.0/, last access: 4 February 2023)

---

## Author Comment (AC2)

This paper is an editorial overview of 10 papers published representing the The Baltic Earth Assessment Reports. Consequently, its content and quality rests on those ten detailed assessments focusing on the Baltic Earth Grand Challenges and working group topics that summarize current knowledge about past, present and future climate changes of the Baltic Sea region. This reviewer also conducted peer reviews of two of those 10 papera and found them very comprehensive and insightfully synthetic. This editorial overview accurately and effectively summarizes the main point of each of the ten papers and presents a discussion of the key knowledge gaps that cut across them. I think it is a very effective capstone for the collection of papers. This Baltic Earth Assessment Report builds on previous regional assessments for the Baltic Sea region in a way that fills an important gap by integrating regional knowledge and scale with the global assessments conducted by the IPCC. In that regard it is unique among large coastal and ocean regions, which would do well to emulate this example.

I found only a few minor points in the text that the authors may wish to consider in finalizing the paper, indicated below by line numbers:

Thank you very much for your positive and helpful comments. We have implemented all requested changes. Please find our detailed response in red below.

32 I am surprised that Germany is not included in this list. Although only a small part of Germany lies in the catchment, this is about the same area as Denmark and Germany is the only littoral state not included.

Correct. We have added Germany.

36 "Anthroposphere" is not a term yet in very widespread usage, but is mentioned several times in this summary. It would helpful to provide a more expansive definition here when it is first mentioned, something like "that part of the environment that is made or modified by humans for use in human activities."

We agree and added your definition.

82 "Rural" generally refers to countryside, typically including agriculture, would it be better to state this as "subarctic wilderness in the north"?

We agree and replaced "rural".

157–158 Awkwardly stated: "the increase of anoxic bottoms have still increased . . ." Do you mean the area of anoxic bottoms?

Corrected.

186 It is not exactly clear what is meant by "available climate model ensembles are too small"? Too few, sparse, limited?

You are right. Following the suggestion by Jouni Räisänen, we rephrased the sentence "suggesting that these changes will likely be small compared with natural variability".

197–198 Should not the acceleration of sea level rise in the Baltic be somewhat more that the global mean because of its greater sensitivity to gravitational contributions from ice loss in Antarctica?

Surprisingly, the Baltic sea level rise is projected to be somewhat smaller than the global mean sea level rise although uncertainties are large. From Meier et al. (2022b). "For the period 2090–2099, relative to 1980–1999, and based on the SRES A1B scenario, the projected absolute sea level rise in

the Baltic Sea was estimated to be about 80% of the global increase (Grinsted et al., 2015). These results were confirmed by other studies for other scenarios and slightly different reference periods (e.g. Kopp et al., 2014; Grinsted, 2015) and summarised by Pellikka et al. (2020) who, for the period 2000–2100, documented an ensemble mean absolute sea level rise in the Baltic Sea of about 87% of the global mean sea level rise.

Please find the revised manuscript below.

Thank you very much!

Markus on behalf of the co-authors

**The Baltic Earth Assessment Reports**

H. E. Markus Meier[1], Marcus Reckermann[2], Joakim Langner[3], Ben Smith[4,5] and Ira Didenkulova[6]

[1]Department of Physical Oceanography and Instrumentation, Leibniz Institute for Baltic Sea Research Warnemünde, Rostock, Germany
[2]International Baltic Earth Secretariat, Helmholtz-Zentrum Hereon, Geesthacht, Germany
[3]Swedish Meteorological and Hydrological Institute, Norrköping, Sweden
[4]Hawkesbury Institute for the Environment, Western Sydney University, Australia
[5]Department of Physical Geography and Ecosystem Science, Lund University, Sweden
[6]Department of Mathematics, University of Oslo, Norway

*Correspondence to*: H. E. Markus Meier (markus.meier@io-warnemuende.de)

**Abstract.** Baltic Earth is an independent research network of scientists from all Baltic Sea countries that promotes regional Earth system research. Within the framework of this network, the Baltic Earth Assessment Reports (BEARs) were produced in the period 2019-2022. These are a collection of 10 review articles summarising current knowledge on the environmental and climatic state of the Earth system in the Baltic Sea region and its changes in the past (palaeoclimate), present (historical period with instrumental observations) and prospective future (until 2100) caused by natural variability, climate change and other human activities. The division of topics between articles follows the grand challenges and selected themes of the Baltic Earth Science Plan, such as the regional water, biogeochemical and carbon cycles, extremes and natural hazards, sea level dynamics and coastal erosion, marine ecosystems, coupled Earth system models, scenario simulations for the regional atmosphere and the Baltic Sea, and climate change and impacts of human use. Each review article contains an introduction, the current state of knowledge, knowledge gaps, conclusions and key statements, based on which recommendations are made for future research. In parallel, Baltic Earth's ongoing outreach work has led to the publication of an information leaflet on climate change in the Baltic Sea, which has been published in two languages so far, and the organisation of stakeholder conferences and workshops.

**1 Introduction**

**1.1 BALTEX/Baltic Earth history**

Baltic Earth is an international research network dealing with Earth system science of the Baltic Sea region (https://baltic.earth, last access: 4 February 2023). The catchment area of the Baltic Sea is about four times larger than the Baltic Sea surface and is part of mainly the countries Belarus, Denmark, Estonia, Finland, Germany, Latvia, Lithuania, Poland, Russia, and Sweden (Fig. 1). Baltic Earth is politically independent and focusses on research on the water and energy cycles, climate variability and climate change, water management and extreme events, and related impacts on marine and terrestrial biogeochemical cycles. Human impact on the Earth system in more general terms, i.e. the anthroposphere, defined as the part of the environment that is made or modified by humans for use in human activities, was added to the 2017 Baltic Earth Science Plan (https://baltic.earth/grandchallenges, last access: 4 February 2023).

Baltic Earth is the successor of the Baltic Sea Experiment (BALTEX) programme, which was founded in 1993 as a GEWEX continental-scale experiment (Global Energy and Water Exchanges, a core project of the World Climate Research Programme) (Reckermann et al., 2011). During Phase I (1993–2002), BALTEX was primarily devoted to hydrological, meteorological and oceanographic processes in the Baltic Sea drainage basin, hence focussed on physical aspects of the Earth system. During the second phase (Phase II: 2003–2012), the programme was expanded to encompass regional climate research, carbon and biogeochemical cycles, engagement with stakeholders and decision makers via assessment reports, as well as communication and education, i.e. organizing summer and winter schools and international master courses.

In 2013, Baltic Earth was launched with a novel science plan reinforcing efforts to address Grand Challenges on (1) salinity dynamics in the Baltic Sea, (2) land-sea biogeochemical linkages, (3) natural hazards and extreme events, (4) sea level and coastal dynamics, (5) regional variability of water and energy exchanges, and (6) multiple

drivers of regional Earth system changes (Meier et al., 2014). Working groups were initiated on coupled Earth system models, the Baltic Sea Model Intercomparison Project (BMIP), education, outreach and communication, and scenario simulations for the Baltic Sea.

Baltic Earth and its predecessor BALTEX have produced three extensive regional assessment reports since 2008. The first two (The BACC Author Team, 2008, and The BACC II Author Team, 2015) emphasised climate change and its impacts in the Baltic Sea region and were published as text books, while the third, the Baltic Earth Assessment Reports (BEARs), was published in the format of a special issue in *Earth System Dynamics*, in 2022. The BEARs and BACC assessment reports fill a gap relative to the assessment reports of the Intergovernmental Panel on Climate Change (IPCC), since the latter focus on global scales, and do not provide detailed local to regional information about the current state of knowledge on climate change and its impacts in the Baltic Sea region. The BEARs provide a comprehensive and up-to-date overview of the state-of-the-art research on the compartments of the Earth system in the Baltic Sea region encompassing processes in the atmosphere, on land and in the sea, including the marine and terrestrial ecosystems as well as processes and impacts related to the anthroposphere.

The BEARs wrap together the currently available published scientific knowledge, updating the second assessment report of climate change in the Baltic Sea basin (The BACC II Author Team, 2015) based on the scientific literature. The present BEAR Special Issue comprises 10 articles on the Baltic Earth Grand Challenges and working group topics including a summary of the current knowledge about past, present, and future climate changes for the Baltic Sea region. The articles encompass contributions by 109 authors from 14 countries and reference 2822 scientific articles and institutional reports in their synthesis effort.

**1.2 Baltic Sea region characteristics**

The Baltic Sea is a semi-enclosed, shallow sea with limited water exchange with the world ocean and small tidal amplitudes. Situated in Northern Europe, the climate of the region is highly variable because it is located in the transition zone between maritime and continental climates, influenced by the North Atlantic and Arctic regions. The river discharge from the large catchment area causes a pronounced gradient in sea surface salinity from about 20 g kg$^{-1}$ in the Danish Straits region to about 2 g kg$^{-1}$ or even less in the northern and eastern reaches of the Baltic Sea. Hence, the Baltic Sea is brackish, with habitats of maritime species in the south-west and freshwater species in the north-east. The Baltic Sea catchment area covers an area of almost 20% of the European continent. 85 million people in 14 countries live in the catchment area, which stretches from the temperate, densely populated south to the subarctic wilderness in the north.

Episodically, large amounts of saltwater from the North Sea cross the sills, located in the Danish Straits, into the Baltic Sea and ventilate the Baltic Sea deepwater. These events require a period of about 20 days of easterly winds that lower the Baltic sea level, followed by a period of about equal duration of strong westerly winds that push saltwater into the Baltic Sea. These events are called Major Baltic Inflows (MBIs) and are important for the water exchange between the North Sea and the Baltic Sea. Mixing is low compared to other seas, with an origin at the lateral boundaries, because tidal amplitudes are very small and energetically unimportant.

In recent decades, environmental conditions in the Baltic Sea have changed considerably. For instance, since 1980 the Baltic Sea has been warming more than any other coastal sea (Fig. 2), causing shorter sea ice and snow covers over land during winter. Furthermore, rising nutrient inputs from land in the 1950s/60s, caused by population growth accompanied by sewage water release to the Baltic Sea and intensified usage of fertilisers from agriculture, led to eutrophication and spreading of hypoxic and anoxic areas. Since the 1980s, nutrient input into the Baltic Sea has been steadily decreasing, but this has not yet led to a significant improvement in oxygen conditions. Recent trends in acidification are smaller than in the world ocean, in particular in the northern Baltic Sea, because positive trends in alkalinity supply counteract the acidification.

**2 Methods**

Following the BACC[1] Author Team (2008) and the BACC II Author Team (2015), the BEAR project is an effort to summarise scientific knowledge about climate change and other drivers of Earth system changes and their impacts on the Baltic Sea region. The two BACC books adopted a format inspired by the IPCC assessment reports. The present special issue in Earth System Dynamics is the third assessment. It takes a new format of Baltic Earth Assessment Reports (BEARs), encompassing 10 peer-reviewed scientific journal articles. The assessed knowledge was extracted from the scientific literature such as peer-reviewed articles, reports from research institutions, and published datasets. Importantly, literature from non-governmental organisations with a political or economic interest, political parties and other stakeholder organisations was excluded from the assessment, ensuring that only scientific knowledge informed the assessment. The BEARs focus on publications after 2012/2013, the year of the editorial deadline of the second assessment report. Whenever possible, uncertainty levels of the BEAR results are classified based on a matrix of consensus within the scientific literature and the documented evidence of detected changes and their attributed drivers such as climate change and human use. For a high confidence of a certain statement, high levels of both scientific consensus and evidence cases are required. Instances of disagreement and knowledge gaps are documented and discussed, informing priorities for future research.

Together with the intergovernmental Baltic Marine Environment Protection Commission (HELCOM), Baltic Earth formed an Expert Network on Climate Change (EN CLIME). The aim of the expert network is to regularly produce a climate change fact sheet (CCFS, 2021[2]) from the BEAR material. In 2021, it was published for the first time (http://helcom.fi/ccfs, last access: 4 February 2023). The CCFS contains some background information, a map showing regional future climate changes for selected parameters under the greenhouse gas concentration scenario RCP4.5 and information about 34 variables, directly and indirectly affected by climate change. For each parameter, a general description, past and prospective future changes, other drivers than climate change (only for the indirect parameters), knowledge gaps, policy relevance and references are presented. More than 100 scientists contributed to the compilation of the fact sheet which was coordinated by the HELCOM secretariat. Updated versions are planned for intervals of seven years. Like the BEARs, the fact sheet was peer-reviewed and quality assured. It has
* * *
[1] Assessment of Climate Change in the Baltic Sea Basin (BACC); https://baltic.earth/bacc, last access: 4 February 2023

[2] https://helcom.fi/wp-content/uploads/2021/09/Baltic-Sea-Climate-Change-Fact-Sheet-2021.pdf, last access: 4 February 2023

so far been translated to German (Ostsee Klimawandel Faktenblatt, 2022[3]) and translations to other languages are planned, enhancing accessibility to stakeholders.

In this editorial, we highlight the main findings and knowledge gaps as detailed by the BEARs and future work is proposed.

**3 Results**

[revised manuscript text omitted]

---

## Author Comment (AC3)

**General comments:**

By wrapping up and summarising a series of Baltic Earth Assessment Reports (BEARs) already published in ESD, this article definitely corresponds to the scope of journal. While the article is in a way "a review of reviews", its novelty is mostly expressed in the Discussion part proposing and justifying further research directions to be addressed by Baltic Earth. Although the presented conclusions are fully supported by the material presented, it would be advisable to supplement the concluding part with a brief outlook on expected future continuation/update of the series. Throughout the paper authors emphasize the review character of the reports it summarises, emphasizing impressive amount of reviewed scientific literature. Content of this paper is clearly reflected in tis title. The list of references is fully adequate both in terms of number and quality.

In several occasions it is felt that there is a space for improving language (e.g. word order in sentences), therefore prior final publishing it is advisable to carry out a repeated language check.

Thank you very much for your positive and helpful comments. We have implemented most of the requested changes. A final language check was performed. Please find our detailed response in red below.

**Specific comments:**

L27: consider completing the Abstract with mentioning contribution of BEARs to producing HELCOM's CCFS – this is very important input to practice.

We modified the abstract as suggested.

L31-32: consider relocating this sentence to section 1.2.

We moved the sentence to section 1.2 as suggested.

L44: consider replacing "carbon and biogeochemical cycles" by "biogeochemical cycles including carbon". Kulinski et al. , 2022 encompasses both nutrients and carbon.

Done as suggested

L181-183: sentence is unclear

For clarity, we revised the sentence: "Changes in relative sea level extremes will depend on the competing impacts of the rising global mean sea level, the gravitational effect of the melting of the Greenland and Antarctic ice sheets, changes in wind fields, etc. and the regionally differing Glacial Isostatic Adjustment (GIA) resulting in land uplift or subsidence."

L208-212: consider rephrasing. Listing all phenomena, human activities, pressures etc. "factors" might be seen a confusing. Note that "land cover" is mostly natural while "land use" completely human-induced.

In Reckermann et al. (2022), "land use and land cover" are considered as one "factor" modified by human activities. Hence, we prefer the original classification. No changes were performed.

L215-217: Clarify the sentence starting with "After climate change..."

For clarity, we slightly modified the sentence: "After climate change, shipping and land use/agriculture have the strongest impacts on the other factors, while fisheries, marine ecosystems and agriculture are the most affected."

L301-302: seem repetition of L205-219

The sentence refers to the review article by Meier et al. (2022b). For clarity, we added this reference to the sentence: "The anthroposphere is not part of this assessment by Meier et al. (2022b) but instead is discussed in detail by Reckermann et al. (2022)."

L333-343: consider shortening. Excessive list of all new data sources draws attention away from the main point: necessity to model small scale processes

We prefer to keep the list of novel observing systems to illustrate which observations are being discussed. The list is specific for the Baltic Sea.

L370-371: consider clarifying the point. E.g. Development of integrated Earth system models accounting for anthropogenic changes...

We modified the sentence as suggested.

L381-383: Clarify sentence

The sentence was rephrased and should now be better understandable: "Of course, the factors discussed by Reckermann et al. (2022) cannot exhaustively take into account the whole Earth system and all interactions, and there is a bias in the selection of factors towards marine-related parameters and activities."

L403: competing the Concluding remarks by some future outlook on BEARs would be strongly advisable

We agree and added the following outlook: "Since the information summarized by the BEARs are used extensively in science and management, it is recommended that a new update of the reports will be conducted in about seven years."

Figure 3: Picture is difficult to comprehend. Some links seem missing. Consider revisiting.

The causal relationships between the Earth system parameters under consideration are subjective. However, please note that only the links of prevailing attribution to climate change with sufficiently high confidence (according to the IPCC definition) are shown by arrows. These relationships are the result of an assessment and thoroughly explained in the review study by Meier et al. (2022b). Hence, we will not change the original figure adopted from Meier et al. (2022b). For more detailed explanations, the reader is referred to Meier et al. (2022b).

Table 3. Categorising economic sectors into "natural", "industrial" and "services" and attributing specific "factors" to these "sectors" (e.g. ecosystem services) seems confusing. Any chance use more traditional approach of categorising pressures? e.g. https://www.eionet.europa.eu/etcs/etc-icm/products/etc-icm-reports/etc-icm-report-4-2019-multiple-pressures-and-their-combined-effects-in-europes-seas

The "factors" discussed by Reckermann et al. (2022) are not only pressures but a mixture of drivers, pressures, states, impacts and responses. Table 3 is a proposal by the authors how to separate drivers/pressures and state variables to simulate with an integrated Earth system model the impacts with the aim to propose responses. As there are other approaches, we added a sentence with the proposed reference by Korpinen et al. (2019).

**Technical corrections:**

L19: end: consider replacing "between" by "among"

Done.

L43: delete "Phase II" in brackets

Done.

L59: consider placing BACC before BEARs i.e. in chronological order.

Done.

L65: consider deleting repeated "anthroposphere"

We have not deleted "anthroposphere" because there is no repetition.

L67-69: consider reformulating this sentence to clarify that BOTH BACC books and Bears are based on scientific literature.

For clarification, we added "the latest scientific literature".

L71-72: consider leaving out "in their synthesis effort"

Done.

L103: replace "assessment of a new format" by "assessment in a new format"

Based upon a comment by another reviewer, the sentence was rephrased.

L117-118: consider replacing "from the Bear material" by "based on the BEAR material"

Done.

L119: move http address of CCFS to footnote

Done.

L129-130: consider improving language: ...we highlight and ... propose...

Done.

L132-133: Seems a repetition of L69-71

Correct. We deleted the sentence and added the reference to the Baltic Earth Science Plan (2017) to the introduction.

L147: consider relacing "In contrast" by "In supplement"

We replaced "In contrast" with "As an advance over"

L226: consider replacing "brightening" by "reducing cloud cover"

We prefer to keep the scientific well-defined term "recent brightening" because also the cloud composition might have changed.

L231: "river input of dissolved..."

Done.

L232: consider replacing "proceeds" by "succeeds" or continues

Done.

L328-329: consider modifying word order

We rephrased the first sentence of the discussion: "One of the main objectives of the BEAR project was to identify knowledge gaps in the Earth system science of the Baltic Sea region so that these can be further addressed in future research."

L453: consider finding better phrase for "corresponding arguments"

We rephrased the sentence "Similar arguments …."

L377: consider streamlining phrase "…in many cases several causes are responsible…"

We rephrased the sentence: "Such a holistic view is urgently needed, as in many cases several reasons are responsible for the observed changes in the Earth system and attributing them to only one factor, e.g. climate change, would be an inadmissible simplification."

L395-396: consider simplifying the last sentence of this para

[revised manuscript text omitted]